# AGENT*: OPTIMIZING TEST-TIME COMPUTE FOR MULTI-AGENT SYSTEMS WITH MODULARIZED COLLABORATION

## ABSTRACT

Scaling test-time computation improves large language model performance without additional training. Recent work demonstrates that techniques such as repeated sampling, self-verification, and self-reflection can significantly enhance task success by allocating more inference-time compute. However, applying these techniques across multiple agents in a multi-agent system is difficult: there does not exist principled mechanisms to allocate compute to foster collaboration among agents, to extend test-time scaling to collaborative interactions, or to distribute compute across agents under explicit budget constraints. To address this gap, we propose AGENT*, a framework for *optimizing* test-time compute allocation in multi-agent systems under fixed budgets. AGENT* introduces *modularized collaboration*, formalized as callable functions that encapsulate reusable multi-agent workflows. These modules are automatically derived through self-play reflection by abstracting recurring interaction patterns from past trajectories. Building on these modules, AGENT* employs *a dual-level planning architecture* that optimizes compute allocation by reasoning over the current task state while also *speculating* on future steps. Experiments on complex agent benchmarks demonstrate that AGENT* consistently outperforms baselines across diverse budget settings, validating its effectiveness for multi-agent collaboration in inference-time optimization.

## 1 INTRODUCTION

Increasing inference-time computation (Snell et al., 2024; Muennighoff et al., 2025; Balachandran et al., 2025; Zhang et al., 2025b) has emerged as a powerful strategy for improving the performance of large language model (LLM) without additional training. Recent models, such as OpenAI's o1 and o3 and Anthropic's Claude Sonnet 3.7, have demonstrated strong reasoning capabilities through techniques such as self-correction (Madaan et al., 2023; Chen et al., 2025a), iterative verification (Lee et al., 2025), and best-of-N sampling (Brown et al., 2024). However, extending test-time scaling to multi-agent systems (OpenAI, 2025; Google, 2025; Anthropic, 2024; OpenAI, 2024) introduces unique challenges.

First, current test-time scaling techniques do not extend effectively to multi-agent systems. In single-agent settings, extra compute can be spent directly on repeated sampling, verification, or reflection. In contrast, multi-agent systems face the challenge of deciding how additional compute should be *allocated* across agents and interactions. Existing approaches, such as the widely adopted orchestrator–worker paradigm (Hadfield et al., 2025; Tran et al., 2025), decompose tasks and invokes agents sequentially, but this setup neither facilitates genuine collaboration nor encourages the system to invest extra compute in coordination. As a result, synergies between complementary agents remain underexploited, and execution is biased toward fixed patterns, limiting adaptability to varying task demands.

Second, there is no principled mechanism for maximizing performance under a fixed compute budget. Existing strategies, such as budget enforcement (Muennighoff et al., 2025), typically rely on static rules or heuristics. Such approaches cannot adjust compute allocation across inference steps to account for varying complexities, nor can they anticipate future compute demands. Anticipation

is essential for scaling, because deciding whether to scale compute for the current step depends on estimating how much is likely to be needed later. Without this foresight, systems risk overspending early or under-investing when tasks become more challenging. This limitation is particularly acute in multi-agent systems, where compute must be strategically distributed across agents with heterogeneous capabilities and costs.

In this paper, we address the core question: *How can multi-agent systems maximize performance under fixed compute budgets while leveraging complementary agents capable of collaboration?* To answer this, we propose AGENT*, a general framework for budget-constrained optimization of test-time compute in multi-agent systems. At its core, AGENT* introduces the abstraction of *collaboration modules* — modular, callable functions that encapsulate high-level coordination strategies among agents. Each module defines a structured multi-agent workflow with a standardized input schema and functionality, making agent interactions composable and reusable, similar to tools. Collaboration modules provide a principled way to allocate and scale compute effectively, since the orchestrator can decide when and how to invoke more *compute-intensive forms* of collaboration depending on task state and budget. In this view, multi-agent collaboration reduces to a function-calling problem, with a meta-agent dynamically choosing which module to invoke. The modules themselves are automatically induced through self-play reflection on past trajectories, ensuring they capture reusable coordination patterns without manual design.

Having established reusable collaboration modules, the central challenge becomes how to allocate computation across them under budget constraints. To address this, AGENT* employs a *dual-level planning architecture* that integrates short-horizon and long-horizon planning. The short-horizon planning proposes candidate next actions, either invoking a collaboration module or an individual agent, conditioned on the current task state. In parallel, the long-horizon planning, inspired by speculative decoding (Leviathan et al., 2023), performs high-level speculation by reasoning abstractly over potential sequences of collaboration modules instead of concretely executing them. This yields low-cost estimates of budget feasibility, enabling the system to reason about when additional compute should be saved or spent. Together, the two levels operate in an A*-like fashion, where the short-horizon planner prioritizes promising near-term actions and the long-horizon planner supplies a lookahead signal that aligns decisions with budget-aware trajectories.

To summarize, this paper makes the following contributions:

- We formulate the problem of optimizing test-time compute allocation for multi-agent systems under fixed budget constraints with a set of agents capable of collaboration.

- We propose AGENT*, a general framework for budget-aware multi-agent collaboration. AGENT* introduces *collaboration modules* to facilitate effective multi-agent collaboration, and employs a *dual-level planning architecture* that balances short-horizon action selection with long-horizon speculation to allocate compute effectively under budget constraints.

- We develop *self-play reflection* to collect cost estimates of agent and collaboration module executions and automatically induce collaboration modules from recurring interaction patterns.

- We demonstrate that AGENT* consistently outperforms baselines across challenging multi-agent benchmarks, achieving higher task success rates and more effective budget utilization.

## 2 ORCHESTRATOR–WORKER FRAMEWORK UNDER BUDGET CONSTRAINTS

In this section, we introduce the general setting of orchestrator–worker framework (Hadfield et al., 2025; Tran et al., 2025) in multi-agent systems that operate under a fixed test-time compute budget. The goal is to solve an input task while ensuring total computation stays within budget.

Let $\mathcal{A} = \{a_1, \ldots, a_N\}$ denote the set of available *worker agents*, each initialized with a large language model (LLM) with distinct capabilities and an associated cost of invocation. A task is solved through a sequence of intermediate states $\{s_t\}_{t=0}^{H}$, where $s_0$ is the initial task description and $s_H$ is the final output returned by the system. At each step $t$, the orchestrator selects an *action* $\alpha_t$

$$\alpha_t = (a_t, \upsilon_t), \quad a_t \in \mathcal{A},$$

where $\upsilon_t$ specifies the subtask or input to the chosen worker. Executing $\alpha_t$ produces an output $o_t$, and the new state is defined as $s_t = (\upsilon_t, o_t)$. Each action incurs a cost $\text{cost}(\alpha_t)$, and the cumulative

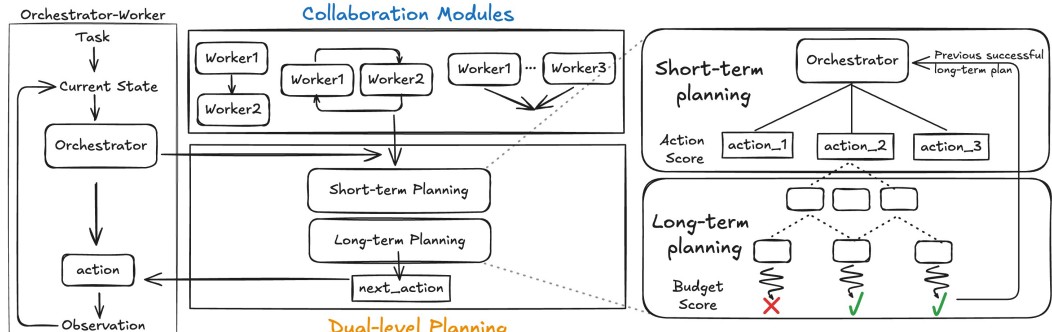

Figure 1: Overview of AGENT*. The framework extends the standard orchestrator–worker paradigm by introducing *collaboration modules*, which encapsulate reusable multi-agent workflows, and a *dual-level planning architecture*, which integrates short-term and long-term planning to select the most promising next action.

cost is constrained by the budget

$$\sum_{t=1}^{H} \text{cost}(\alpha_t) \leq B.$$

The system's behavior at step $t$ is governed by a policy $\pi$ that selects the next action based on the execution history $\mathcal{H}_t = \{ s_r \}_{r=0}^{t-1}$.

We instantiate the policy $\pi$ with a LLM (Yao et al., 2020; Huang et al., 2022; Yao et al., 2023), which serves as the backbone for the orchestrator agent. Following the ReAct framework (Yao et al., 2023), the orchestrator generates subtasks, assigns them to worker agents, evaluates their outputs, and determines subsequent actions until a final solution is produced or the budget is exhausted.

## 3 AGENT*

Multi-agent systems built on the orchestrator–worker paradigm (Hadfield et al., 2025; Tran et al., 2025) suffer from limited coordination and inefficient use of additional compute. Synergies between agents are underexploited, and budget-aware optimization has not been well studied. To address these issues, we propose AGENT*, which extends the orchestrator–worker framework with two key components.

At the core of AGENT* are collaboration modules (§3.1), that are modular functions encapsulating reusable coordination strategies and transforming multi-agent interaction into a structured function-calling problem. Building on this abstraction, AGENT* employs a dual-level planning architecture (§3.2) that balances short-horizon action selection with long-horizon speculation, enabling adaptive and forward-looking compute allocation under strict budget limits. An overview is shown in Figure 1.

### 3.1 COLLABORATION MODULES

The traditional orchestrator–worker framework is constrained by sequential, one-agent-at-a-time execution. This design becomes inefficient on complex, open-ended tasks, as it is difficult to capture synergies between agents with complementary capabilities. In particular, the sequential setup limits coordination, since intermediate results are not jointly integrated across agents, leaving subtasks only partially addressed. Moreover, this rigid design restricts how additional compute can be used: allocating more resources to a single agent, through repeated sampling or iterative verification, merely amplifies that agent's behavior without fostering cross-agent collaboration. Unlike test-time scaling in single-agent LLMs, such additional compute does not provide a principled path to improving performance in multi-agent systems.

To address these limitations, we introduce *collaboration modules* — modular, callable functions that encapsulate high-level coordination strategies among multiple agents. Each module specifies

a standardized and reusable workflow, such as combining outputs from multiple agents or chaining agents in a pipeline, and can be invoked with a single function call. This abstraction reframes multi-agent collaboration as a function-calling problem, where the orchestrator decides which collaboration module to invoke at each step. By structuring interactions in this way, collaboration modules enable richer coordination and provides a scalable and principled mechanism for utilizing test-time compute more effectively, as additional budget is allocated to cross-agent collaboration rather than simply amplifying the behavior of individual agents.

Formally, we define a collaboration module as

$$m = (\mathcal{S}, \kappa), \quad \mathcal{S} \subseteq \mathcal{A},$$

where $\mathcal{S}$ denotes a subset of worker agents and $\kappa$ is a coordination strategy that specifies how these agents interact. The action space available to the orchestrator is therefore expanded to include not only the individual base agents but also collaboration modules:

$$\mathcal{A}' = \mathcal{A} \cup \mathcal{M},$$

where $\mathcal{A}$ is the set of base worker agents and $\mathcal{M}$ is the set of collaboration modules. From this point onward, we assume that $a_t$ in action $\alpha_t$ is drawn from $\mathcal{A}'$, enabling the orchestrator to invoke either a single agent or a collaboration module at each step.

### 3.2 Dual-Level Planning Architecture

While collaboration modules provide a structured abstraction for multi-agent coordination, they also introduce the challenge of how to allocate computation effectively under budget constraints. Distributing test-time resources across modules is inherently uncertain, since the system cannot know in advance which modules will be invoked or how their interactions will unfold. This uncertainty motivates the need for a planning mechanism that adaptively determines which modules to invoke and how to allocate computation across them, ensuring resources are used efficiently to maximize task performance under strict budget constraints.

Inspired by traditional A* search (Hart et al., 1968; Meng et al., 2024), AGENT* frames planning as the exploration of a search tree, where each node represents a decision point at step $t$ for invoking an action $\alpha_t = (a_t, \upsilon_t)$, evaluated by the cumulative gain $g(\alpha_t)$ and the future gain $h(\alpha_t)$[1]. Analogously, in AGENT*, short-term planning plays the role of $g(\alpha_t)$: it expands candidate next actions based on the policy $\pi(\cdot \mid \mathcal{H}_t)$ and assigns each a score reflecting the short-term gain. Long-term planning corresponds to $h(\alpha_t)$: it speculates over possible future trajectories to estimate whether subsequent steps will remain feasible under the budget. By combining these two levels, the orchestrator effectively selects the candidate with the highest overall utility score $f(\alpha_t) = g(\alpha_t) + h(\alpha_t)$, ensuring that immediate actions are consistent with budget-aware long-term feasibility.

#### 3.2.1 Short-Term Planning

In short-term planning, AGENT* evaluates the immediate utility of a candidate action $\alpha_t = (a_t, \upsilon_t)$ proposed by the policy $\pi$. At each step $t$, the policy generates $K$ candidate actions by drawing from the distribution

$$\mathcal{C}_t = \{\alpha_t^{(1)}, \ldots, \alpha_t^{(K)}\} \sim \pi(\cdot \mid \mathcal{H}_t, \mathcal{T}_{\text{feasible},t-1}),$$

where $\mathcal{T}_{\text{feasible},t-1}$ denotes the set of budget-feasible speculative trajectories carried over from the previous step. This conditioning ensures that short-term proposals are informed by speculated budget-feasible high-level plans rather than sampled solely from $\mathcal{H}_t$. As a result, the candidate set reflects not only the current task context but also long-term budget feasibility, yielding proposals that are both contextually grounded and more likely to lead to successful completions within budget.

To assess these candidates, we compute a self-consistency score (Wang et al., 2022), which serves as a proxy for their effectiveness. Formally, let $\phi((a, \upsilon)) = a$ extract the agent or module from an action. The short-term gain is defined as

$$g\left(\alpha_t^{(i)}\right) = \frac{1}{K} \sum_{k=1}^{K} \mathbf{1}\left[\phi\left(\alpha_t^{(k)}\right) = \phi\left(\alpha_t^{(i)}\right)\right],$$

---

[1]Here, cumulative and future gains are interpreted as the inverse of cumulative and future costs in A*.

where $\mathbf{1}[\cdot]$ is the indicator function. Actions with higher self-consistency receive larger $g(\alpha_t^{(i)})$ values, indicating stronger evidence that they will contribute effectively to task progress. This mechanism provides a lightweight yet reliable signal for prioritizing near-term actions before long-term feasibility is considered.

### 3.2.2 LONG-TERM PLANNING

While short-term planning evaluates the immediate promise of an action, long-term planning speculates on its feasibility under the remaining budget. The goal is to compute a budget feasibility score $h(\alpha_t^{(i)})$ that captures whether choosing $\alpha_t^{(i)}$ as the next action is likely to keep future trajectories within budget, without actually executing $\alpha_t^{(i)}$ and following actions.

Concretely, the orchestrator agent expands each candidate action into *speculative trajectories*, which are abstract rollouts representing possible continuations of agents or collaboration modules. These trajectories are generated symbolically at the module level, so no agents are invoked, keeping the procedural lightweight. Each trajectory is associated with an estimated cumulative cost, and any that exceed the remaining budget are filtered out. Formally, let $\mathcal{T}_t(\alpha_t^{(i)})$ denote the set of speculative trajectories beginning with $\alpha_t^{(i)}$, and let $\text{cost}(\tau)$ represent the estimated cumulative cost of a trajectory $\tau \in \mathcal{T}_t(\alpha_t^{(i)})$. Given the remaining budget $B_t$, we define the set of feasible trajectories as

$$\mathcal{T}_{\text{feasible},t}(\alpha_t^{(i)}) = \{\tau \in \mathcal{T}_t(\alpha_t^{(i)}) \mid \text{cost}(\tau) \le B_t\},$$

so that trajectories whose costs exceed $B_t$ are filtered out. Among the feasible trajectories, we then normalize across the $K$ candidate actions sampled at step $t$. This defines the budget feasibility score:

$$h(\alpha_t^{(i)}) = \frac{|\mathcal{T}_{\text{feasible},t}(\alpha_t^{(i)})|}{\sum_{r=1}^{K} |\mathcal{T}_{\text{feasible},t}(\alpha_t^{(r)})|}.$$

Intuitively, $h(\alpha_t^{(i)})$ reflects the likelihood that an action can be extended into a successful budget-compliant plan. Actions with higher $h(\alpha_t^{(i)})$ values are favored, since they not only appear promising in the short term but are also more likely to sustain progress without violating budget constraints. This speculative lookahead ensures that immediate choices remain consistent with long-term feasibility, complementing the short-term gain $g(\alpha_t^{(i)})$ in the overall utility $f(\alpha_t^{(i)}) = g(\alpha_t^{(i)}) + h(\alpha_t^{(i)})$. Moreover, the resulting feasible speculative set $\mathcal{T}_{\text{feasible},t}$ is carried forward to guide candidate generation in the next step's short-term planning.

Finally, we select the candidate action with the highest utility score as the next action. The dual-level planning procedure is formally presented in Alg. 1.

### 3.3 SELF-PLAY REFLECTION

Dual-level planning requires estimates of the execution cost for both agents and collaboration modules in order to conduct long-term planning. Also, for the collaboration modules, we need effective ways to structure multi-agent collaborations. To address both needs, AGENT* employs *self-play reflection*, an *iterative* process that builds experience by generating execution trajectories on a subset of validation tasks and uses this experience both to compute the average cost of invoking each agent or module and to automatically construct collaboration modules from recurring interaction patterns.

Formally, given the set of actions $\mathcal{A}'$, the system executes multiple trajectories and logs the agents invoked, subtasks addressed, outputs produced, and costs incurred. The average execution cost is then estimated as

$$\widehat{\text{cost}}(a) = \mathbb{E}_{\text{traj},\upsilon}\big[\text{cost}(a,\upsilon)\big], \quad a \in \mathcal{A}',$$

providing the statistics required for long-term planning. While in theory the cost of invoking an agent or module depends on both $a$ and the specific subtask $\upsilon$, enumerating all possible subtasks is infeasible. We therefore relax this assumption and approximate the cost by averaging across observed subtasks in self-play trajectories, treating the resulting estimate as transferable across tasks.

In parallel, successful trajectories are reflected upon by an LLM to identify recurring sequences of agent interactions that consistently contribute to successful task completion. These sequences are

abstracted into reusable collaboration modules, which are added to the action space $\mathcal{A}'$ and refined through further rounds of self-play. Over successive iterations, this process yields a growing library of collaboration modules as well as reliable cost profiles for both modules and base agents.

Through self-play reflection, AGENT* unifies cost estimation and collaboration module discovery in a single data-driven procedure, eliminating the need for manual engineering and enabling more effective budget-optimal planning.

## 4 EXPERIMENTS

To evaluate the effectiveness of AGENT* in multi-agent settings, we conduct experiments on the GAIA and BrowseComp-Plus benchmarks, comparing its performance against baselines under strict budget constraints.

### 4.1 EXPERIMENT SETTINGS

**Dataset.** We evaluate AGENT* on two agent benchmarks: (1) GAIA (Mialon et al., 2023): a real-world QA benchmark that evaluates web browsing and general tool-use ability of language models; and (2) BrowseComp-Plus (Chen et al., 2025b): a benchmark derived from BrowseComp (Wei et al., 2025) that measures the ability for agents to browse the web using a fixed, curated corpus.

**Agents and Collaboration Modules.** For each benchmark, we first instantiate a set of task-specialized agents. In GAIA, we utilize four agents: Search Agent, Browser Agent, Reasoning Agent, and Media Inspector Agent. In BrowseComp-Plus, we use three agents: Retriever Agent, Document Reader Agent, and Critic Agent.

To construct collaboration modules, we conduct five rounds of self-play reflection for each benchmark, with each round producing one candidate module. We utilize the first 30 questions in the benchmark as a validation set to collect cost estimates and collaboration modules. This process yields mainly five distinct modules for GAIA: *interactive search and browse*, *search then browse*, *ensemble search*, *two ensemble reasoning*, and *three ensemble reasoning*. For BrowseComp-Plus, it yields four unique modules: *interactive search*, *ensemble interactive search*, *interactive search then critic*, and *ensemble interactive search then critic*, with the final round producing a duplicate[2].

**Models.** We employ two model families: Claude (Anthropic, 2025) and Qwen3 (Yang et al., 2025). Within the Claude family, we use `Claude-3.7-Sonnet` for the orchestrator, reasoning, critic, and media inspector agents, and `Claude-3.5-Haiku` for the rest. For the Qwen family, we use `Qwen3-32B` for all the agents.

**Budget Constraints.** For the unit of cost and budget, we compute the monetary cost of each action by multiplying its input and output token usage with the official token pricing from AWS Bedrock[3]. We evaluate performance under four budget settings. To determine suitable constraints for each benchmark and model, we first measure the trajectory costs from self-play reflection and take their average as the minimum budget. Larger budgets are then defined either by fixed increments or by exponential scaling. For Claude experiments, we allocate per-question budgets of $0.2, $0.3, $0.4, and $0.5 across both benchmarks. For Qwen experiments, we set budgets of $0.05, $0.1, $0.2, and $0.3 for GAIA, and $0.025, $0.05, $0.1, and $0.2 for BrowseComp-Plus.

**Metrics.** We report results using Acc@$B$, which denotes the accuracy achieved under a budget $B$. This metric reflects the proportion of questions answered correctly while ensuring that the total token-based compute cost does not exceed $B$ for each query. Acc@$B$ enables fair comparison across different compute regimes and highlights how effectively each method converts its allotted budget into correct answers.

---

[2]We provide the prompt for self-play reflection in Figure 3 and detailed descriptions of agents and collaboration modules in Appx. §A

[3]We referred to https://aws.amazon.com/bedrock/pricing/. The actual token price we used is in Appx. §B

| Claude Models | | | | | | | |
|---|---|---|---|---|---|---|---|
| Method | GAIA | | | | BrowseComp-Plus | | | |
| | Acc@0.2 | Acc@0.3 | Acc@0.4 | Acc@0.5 | Acc@0.2 | Acc@0.3 | Acc@0.4 | Acc@0.5 |
| *Fixed Agent Workflow* | | | | | | | | |
| ADAS | 11.72 | 11.72 | 11.72 | 11.72 | 6.20 | 6.20 | 6.20 | 6.20 |
| AFlow | 12.96 | 12.96 | 12.96 | 12.96 | 5.42 | 5.42 | 5.42 | 5.42 |
| *ReAct with test-time scaling* | | | | | | | | |
| ReAct | 35.80 | 35.80 | 35.80 | 35.80 | 24.33 | 24.66 | 24.66 | 24.66 |
|   w/ Best-of-N | 35.80 | 35.80 | 37.03 | 37.65 | 24.33 | 24.80 | 24.80 | 24.80 |
|   w/ Iterative Verification | 35.80 | 36.41 | 38.27 | 37.03 | 24.54 | 25.06 | 25.58 | 24.80 |
| *ReAct with Collaboration Modules* | | | | | | | | |
| Budget-Unaware | 35.80 | 37.65 | 38.27 | 38.27 | 25.32 | 25.50 | 27.08 | 26.20 |
| Budget-Aware Prompt | 36.41 | 41.97 | 43.20 | 43.20 | 24.07 | 26.34 | 27.37 | 28.07 |
| AGENT* | **38.89** | **44.44** | **46.91** | **48.15** | **25.53** | **27.50** | **28.07** | **29.00** |

| Qwen3-32B | | | | | | | |
|---|---|---|---|---|---|---|---|
| Method | GAIA | | | | BrowseComp-Plus | | | |
| | Acc@0.05 | Acc@0.1 | Acc@0.2 | Acc@0.3 | Acc@0.025 | Acc@0.05 | Acc@0.1 | Acc@0.2 |
| *Fixed Agent Workflow* | | | | | | | | |
| ADAS | 4.93 | 4.93 | 4.93 | 4.93 | 2.84 | 2.84 | 2.84 | 2.84 |
| AFlow | 3.70 | 3.70 | 3.70 | 3.70 | 3.35 | 3.35 | 3.35 | 3.35 |
| *ReAct with test-time scaling* | | | | | | | | |
| ReAct | 12.66 | 12.66 | 12.66 | 12.66 | 7.10 | 7.71 | 8.07 | 8.07 |
|   w/ Best-of-N | 12.66 | 13.58 | 13.58 | 14.19 | 7.10 | 7.71 | 7.71 | 8.07 |
|   w/ Iterative Verification | 12.66 | 14.19 | 16.00 | 16.67 | 8.07 | 14.72 | 15.24 | 16.14 |
| *ReAct with Collaboration Modules* | | | | | | | | |
| Budget-Unaware | 15.33 | 16.00 | 16.00 | 16.00 | 10.36 | 16.14 | 17.46 | 17.46 |
| Budget-Aware Prompt | 11.33 | 13.33 | 14.00 | 14.00 | 16.98 | 16.74 | 16.74 | 16.74 |
| AGENT* | **16.00** | **16.67** | **20.00** | **21.33** | **18.07** | **18.92** | **19.42** | **20.72** |

Table 1: Evaluation Results of the baselines using Accuracy under budget (Acc@$B$) on GAIA and BrowseComp-Plus for Claude and Qwen3 model families.

**Baselines.** We consider the following categories of baselines. (1) Fixed Agent Workflow: Baselines with optimized agent workflow that are fixed during the test time. We employ the most representative works, ADAS (Hu et al., 2024) and AFlow (Zhang et al., 2024a). (2) ReAct with test-time scaling: We evaluate on the standard ReAct framework (Yao et al., 2023) along with two well-established test-time scaling methods, Best-of-N (Brown et al., 2024) and Iterative Verification (Madaan et al., 2023). (3) ReAct with Collaboration Modules: We incorporate collaboration modules into the ReAct loop and evaluate two settings: `Budget-Unaware`, where the orchestrator does not receive explicit budget information, and `Budget-Aware Prompt`, where the orchestrator is given explicit budget information in the prompt[4].

## 4.2 BENCHMARK RESULTS

Table 1 presents the experiment results of the baselines on the GAIA and BrowseComp-Plus benchmarks across different budget constraints.

First, fixed agent workflow baselines underperform on both GAIA and BrowseComp-Plus because they cannot adapt agent execution dynamically during inference. These benchmarks contain questions that require highly diverse workflows, causing the optimized workflow in these methods to collapse into a simple, generic pipeline, which makes it difficult for a single, static workflow to address these tasks effectively. Moreover, since these approaches have no mechanism for dynamically allocating compute, their performance remains largely unchanged even as the available budget increases.

Next, ReAct performs noticeably better than fixed agent workflow baselines but exhibits limited utilization of larger budgets. Although adding Best-of-N or Iterative Verification partially mitigates

---

[4] We provide detailed implementation details in Appx. §C

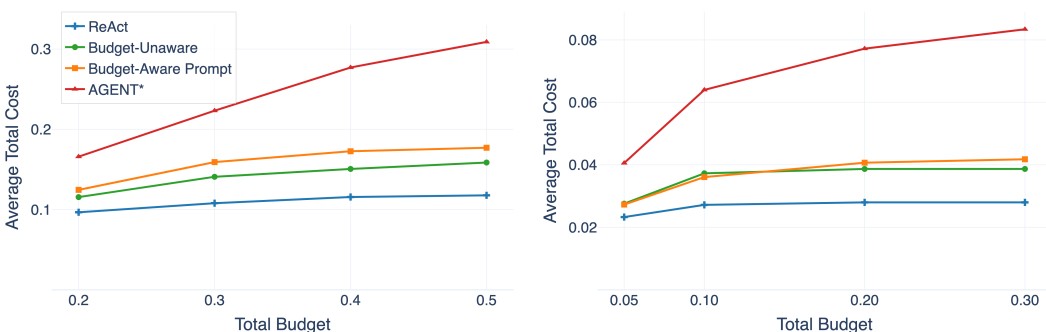

Figure 2: Average total cost vs total budget on GAIA using Claude (left) and Qwen3 (right).

this issue, these methods consume significantly more compute while yielding only minimal and inconsistent accuracy gains, which quickly plateau as the budget increases. This suggests that simply allocating more budget through standard test-time scaling does not guarantee improved performance.

Lastly, collaboration modules lead to clear and consistent improvements. The Budget-Unaware variant already surpasses ReAct by a significant margin, confirming its effectiveness in using test-time compute. Moreover, while Budget-Aware Prompt method occasionally improves accuracy relative to non-budget-aware settings, its effect is inconsistent and in some cases plateaus at higher budgets. AGENT* further achieves the strongest results across all budgets, outperforming both the Budget-Unaware and Budget-Aware Prompt baselines. These gains highlight the value of dual-level planning, which enables the orchestrator to allocate test-time compute more effectively by balancing short-term decisions with long-horizon feasibility.

### 4.3 Cost Utilization under Budget Constraints

We further examine the average total cost incurred when running the GAIA benchmark under different budget constraints, as reported in Figure 2. Across all three baselines, the available budget is consistently underutilized, with the total cost plateauing well below the specified constraint. In contrast, AGENT* achieves substantially higher cost utilization, which transfers into a performance enhancement. This finding highlights a fundamental challenge in multi-agent systems: even when sufficient budget is available, baseline strategies struggle to convert it into effective computation that drives performance gains. By leveraging dual-level planning, AGENT* allocates resources more aggressively when necessary, ensuring that budget headroom is effectively translated into higher performance.

### 4.4 Impact of Short-term and Long-term Planning

To assess the contribution of each planning component in AGENT*, we evaluate them by isolating each component under $0.2 and $0.5 budgets using Claude models on GAIA benchmark. The results are shown in Table 2. Short-term planning alone provides a meaningful boost over vanilla ReAct, as it guides the orchestrator toward more informed local decisions and encourages the use of valuable collaboration modules. Long-term planning alone performs slightly better, particularly at higher budgets, since it enables the model to anticipate down-

| Method | Acc@0.2 | Acc@0.5 |
|---|---|---|
| Only Short-term | 36.41 | 44.44 |
| Only Long-term | 37.65 | 46.91 |
| **Dual-level (Both)** | **38.89** | **48.15** |

Table 2: Evaluation results of short-term and long-term planning on GAIA under $0.2 and $0.5 budget.

stream computation and avoid prematurely committing to suboptimal trajectories. When combined, these complementary behaviors yield the strongest performance, underscoring the value of integrating both local and global planning signals for budget-optimal multi-agent coordination.

## 5 RELATED WORKS

### 5.1 TEST-TIME SCALING FOR LLMS

Existing approaches to increasing test-time compute for LLMs can be broadly categorized into two paradigms (Muennighoff et al., 2025), including parallel scaling and sequential scaling. Parallel scaling methods, such as best-of-N sampling (Brown et al., 2024; Snell et al., 2024), generate multiple candidate solutions in parallel and select the best one using voting, confidence heuristics, or reward models (Wang et al., 2022; Irvine et al., 2023). In contrast, sequential scaling encourages iterative refinement via methods such as chain-of-thought prompting (Wei et al., 2022), self-refinement (Madaan et al., 2023; Chen et al., 2023; Min et al., 2024; Lee et al., 2025), and verifier-guided revision (Gou et al., 2023; Zhang et al., 2024d). While these methods offer promising directions for test-time scaling, they do not consider optimization under a strict compute budget. Recent works have explored budget-constrained settings (Muennighoff et al., 2025; Han et al., 2025; Zheng et al., 2024; Qiu et al., 2025), but they either focus on single LLM setting or target system-level efficiency rather than scaling test-time computation to improve the performance. In contrast, our work focuses on budget-optimal compute allocation in a multi-agent setting, which is not addressed by these prior approaches

### 5.2 DESIGNING AND OPTIMIZING MULTI-AGENT SYSTEM

Multi-agent systems (MAS) have recently gained traction as a way to structure complex tasks through the collaboration of specialized LLM-based agents. Early systems such as CAMEL (Li et al., 2023), AutoGen (Wu et al., 2024), and MetaGPT (Hong et al., 2023) demonstrated the value of explicit role assignment and agent interaction, but relied heavily on manual configurations, including prompt engineering, agent profiling, and fixed communication protocols (Qian et al., 2024). These limitations hinder their adaptability across domains and tasks. In response, recent work has focused on automating various components of MAS design. Some methods treat agent functions as learnable policies (Zhang et al., 2024b;c) or synthesize trajectories for offline agent optimization (Qiao et al., 2024). Others expand the MAS search space to include prompts (Khattab et al., 2023), tools (Zhou et al., 2024), workflows (Li et al., 2024), and reasoning strategies (Shang et al., 2024). DyLAN (Liu et al., 2024) supports dynamic agent composition, while Archon (Saad-Falcon et al., 2024) treats MAS construction as a hyperparameter optimization problem. GPTSwarm (Zhuge et al., 2024) optimizes agent communication using policy gradients, and state-of-the-art systems like ADAS (Hu et al., 2024) and AFlow (Zhang et al., 2024a) perform full workflow optimization using search algorithms or LLM controllers.

More recently, several methods generate workflows on a per-query basis. MaAS (Zhang et al., 2025a) learns distributions over architectures to trade off performance and cost; FlowReasoner (Gao et al., 2025), ScoreFlow (Wang et al., 2025), and Flow (Niu et al., 2025) synthesize or refine workflows using execution feedback, DPO training, or graph-based reasoning. However, these systems aim to produce a single optimized configuration per task and do not support dynamic, inference-time planning under compute budgets, nor do they introduce reusable collaboration abstractions that enable adaptive, budget-efficient multi-agent coordination without retraining or static system design.

## 6 CONCLUSION

We presented AGENT\*, a general framework for budget-constrained optimization of test-time compute in multi-agent systems. AGENT\* leverages collaboration modules as reusable abstractions of multi-agent coordination and employs a dual-level planning architecture that balances short-horizon execution with long-horizon speculation. Extensive experiments demonstrate that AGENT\* consistently outperforms both standard baselines and those augmented with test-time scaling, achieving higher accuracy while utilizing resources more efficiently. These findings underscore the importance of structured collaboration and forward-looking planning for budget-constrained inference, and point toward a promising direction for building more adaptable and compute-efficient multi-agent systems.

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

## A  SUPPLEMENTARY DETAILS ON AGENTS AND COLLABORATION MODULES

For the GAIA benchmark, we employ the following agents:

- *Search Agent*: Given a search query, outputs the google search results
- *Browser Agent*: Given one or more URLs, visits the pages and returns the content of webpages.
- *Reasoning Agent*: Given a problem, performs multi-step reasoning and outputs a final solution.
- *Media Inspector Agent*: Given an image or video link and a question, analyzes the media and answers the question.

In addition, we define the following collaboration modules:

- `interactive_search_and_browse`: A search agent and a browser agent share context and invoke each other interactively.
- `search_then_browse`: First searches for URLs, then browses the retrieved pages for detailed content.
- `ensemble_search`: Generates three distinct queries, spawns three search agents in parallel, and aggregates their results.
- `two_ensemble_reasoning`: Two reasoning agents independently produce reasoning paths, which are aggregated into a final answer.
- `three_ensemble_reasoning`: Three reasoning agents independently produce reasoning paths, which are aggregated into a final answer.

For the BrowseComp-Plus benchmark, we employ the following agents:

- *Retriever Agent*: Given a search query, retrieves the top-5 documents by semantic similarity and returns (`doc_id`, `title`, `snippet`) for each.

- *Document Reader Agent*: Given a `doc_id`, fetches and returns the full document content.
- *Critic Agent*: Given the current task state and available information, identifies missing information and recommends what to search for next.

We define the following collaboration modules:

- `interactive_search`: A search agent and a document reader agent share context and invoke each other interactively.
- `ensemble_interactive_search`: Spawns three `interactive_search` modules in parallel and aggregates their results.
- `interactive_search_then_critic`: First calls the `interactive_search` module, then the critic agent evaluates the gathered information.
- `ensemble_interactive_search_then_critic`: Spawns three `interactive_search_then_critic` modules in parallel and aggregates their results.

## B  TOKEN PRICING FOR COST COMPUTATION

We compute the monetary cost of each action by multiplying its input and output token usage with the official token pricing provided by AWS Bedrock. Table 3 lists the exact token prices used for all models in our experiments. These values correspond to the pricing available at the time of experimentation.

| Model Name | Input Price ($/1K tok) | Output Price ($/1K tok) |
|---|---|---|
| `claude-3-5-haiku-latest` | $0.0008 | $0.004 |
| `claude-3-7-sonnet-latest` | $0.003 | $0.015 |
| `qwen3-32b` | $0.0007 | $0.0028 |

Table 3: Token prices used for calculating costs and budgets.

## C  IMPLEMENTATION DETAILS

Here, we provide detailed implementation details for the baselines. For the best-of-N baseline, we set $N = 3$, but if the budget is exhausted before completing all three attempts, we use the available attempts and apply self-consistency on the answers from the attempts to produce the final answer. For the iterative verification baseline, once an initial answer is generated, any remaining budget is used to prompt the orchestrator to re-examine the trajectory and refine the solution until the budget is fully consumed.

## D  MAIN EXPERIMENT SCORES

We have additionally included the specific performance scores presented in §4.2 in Table 4 and Table 5.

## E  DUAL-LEVEL PLANNING ALGORITHM

We provide algorithm for dual-level planning in Alg. 1.

```
You are given a set of successful execution trajectories produced by
    a multi-agent system.
Each trajectory contains the sequence of agents invoked with a
    subtask, and their intermediate outputs

Your goal is to analyze these trajectories to identify a recurring
    agent interactions that frequently appear in successful
    trajectories.
Such patterns may include:
- Sequential workflows
- Parallel workflows
- Verification or Refinement

For each recurring pattern you identify:
1. Describe the workflow in plain language.
2. Specify the agents involved and their roles.
3. Implement a class that executes this workflow using these agents
4. Explain why this pattern is effective, referring to the trajectory
    evidence.

Here are some examples of the extracted pattern:
{few_shot_demonstrations}

Here are the patterns that are already observed. Avoid outputting
    duplicated patterns.
{collected_collaboration_modules}
```

Figure 3: Prompt used for self-play reflection to induce collaboration modules from successful trajectories.

| Method | GAIA | | BrowseComp-Plus | |
|---|---|---|---|---|
| | Budget ($) | Accuracy (%) | Budget ($) | Accuracy (%) |
| No Module, No Budget Aware | | 35.80 | | 24.33 |
| With Module, No Budget Aware | 0.2 | 35.80 | 0.2 | 25.32 |
| With Module, Budget-Aware | | 36.41 | | 24.07 |
| AGENT* | | **38.89** | | **25.53** |
| No Module, No Budget Aware | | 35.80 | | 24.66 |
| With Module, No Budget Aware | 0.3 | 37.65 | 0.3 | 25.50 |
| With Module, Budget-Aware | | 41.97 | | 26.34 |
| AGENT* | | **44.44** | | **27.50** |
| No Module, No Budget Aware | | 35.80 | | 24.66 |
| With Module, No Budget Aware | 0.4 | 38.27 | 0.4 | 27.08 |
| With Module, Budget-Aware | | 43.20 | | 27.37 |
| AGENT* | | **46.91** | | **28.07** |
| No Module, No Budget Aware | | 35.80 | | 24.66 |
| With Module, No Budget Aware | 0.5 | 38.27 | 0.5 | 26.2 |
| With Module, Budget-Aware | | 43.20 | | 28.07 |
| AGENT* | | **48.15** | | **29.00** |

Table 4: Performance of different methods across different budget on GAIA and BrowseComp-Plus using Claude model family.

| Method | GAIA | | BrowseComp-Plus | |
|---|---|---|---|---|
| | Budget ($) | Accuracy (%) | Budget ($) | Accuracy (%) |
| No Module, No Budget Aware | | 12.66 | | 7.10 |
| With Module, No Budget Aware | 0.05 | 15.33 | 0.025 | 10.36 |
| With Module, Budget-Aware | | 11.33 | | 16.98 |
| AGENT* | | **16.00** | | **18.07** |
| No Module, No Budget Aware | | 12.66 | | 7.71 |
| With Module, No Budget Aware | 0.1 | 16.0 | 0.05 | 16.14 |
| With Module, Budget-Aware | | 13.33 | | 16.74 |
| AGENT* | | **16.67** | | **18.92** |
| No Module, No Budget Aware | | 12.66 | | 8.07 |
| With Module, No Budget Aware | 0.2 | 16.0 | 0.1 | 17.46 |
| With Module, Budget-Aware | | 14.00 | | 16.74 |
| AGENT* | | **20.00** | | **19.42** |
| No Module, No Budget Aware | | 12.66 | | 8.07 |
| With Module, No Budget Aware | 0.3 | 16.0 | 0.2 | 17.46 |
| With Module, Budget-Aware | | 14.00 | | 16.74 |
| AGENT* | | **21.33** | | **20.72** |

Table 5: Performance of different methods across different budget on GAIA and BrowseComp-Plus using Qwen3 model family.

**Algorithm 1:** Dual-Level Planning

**Input:** Policy $\pi$, set of agents/modules $\mathcal{A}'$, initial history $\mathcal{H}_0$, total budget $B$, max steps $T_{\max}$
**Output:** Final output and execution trace

1 $t \leftarrow 0$;
2 $\mathcal{H}_t \leftarrow \mathcal{H}_0$;
3 $B_t \leftarrow B$;
4 $\mathcal{T}_{\text{feasible},0} \leftarrow \emptyset$;
5 **while** $t < T_{\max}$ **and** $B_t > 0$ **and not** SOLVED($\mathcal{H}_t$) **do**
6     Sample candidate actions

$$\mathcal{C}_t = \{\alpha_t^{(1)}, \ldots, \alpha_t^{(K)}\} \sim \pi(\cdot \mid \mathcal{H}_t, \mathcal{T}_{\text{feasible},t-1})$$

    `// Let` $\phi((a,v)) = a$ `extract the agent/module from an action`

7     **for** $i = 1$ **to** $K$ **do**
8

$$g\left(\alpha_t^{(i)}\right) = \frac{1}{K} \sum_{k=1}^{K} \mathbf{1}\left[\phi\left(\alpha_t^{(k)}\right) = \phi\left(\alpha_t^{(i)}\right)\right]$$

9     **for** $i = 1$ **to** $K$ **do**
10         Generate speculative trajectories $\mathcal{T}_t(\alpha_t^{(i)})$ beginning with $\alpha_t^{(i)}$;
11         Define feasible set:

$$\mathcal{T}_{\text{feasible},t}(\alpha_t^{(i)}) = \left\{\tau \in \mathcal{T}_t(\alpha_t^{(i)}) \mid \text{cost}(\tau) \leq B_t\right\}$$

$$h\left(\alpha_t^{(i)}\right) = \frac{\left|\mathcal{T}_{\text{feasible},t}(\alpha_t^{(i)})\right|}{\sum_{r=1}^{K} \left|\mathcal{T}_{\text{feasible},t}(\alpha_t^{(r)})\right|}$$

12     Compute $f(\alpha_t^{(i)}) = g(\alpha_t^{(i)}) + h(\alpha_t^{(i)})$ for $i = 1, \ldots, K$;
13

$$\alpha_t^{\star} = \arg\max_{\alpha \in \mathcal{C}_t} f(\alpha)$$

14     Execute $\alpha_t^{\star} = (a_t^{\star}, v_t^{\star})$ to obtain output $o_t$;
15     Update state: $\mathcal{H}_{t+1} \leftarrow \mathcal{H}_t \cup \{(v_t^{\star}, o_t)\}$;
16     Update budget: $B_{t+1} \leftarrow B_t - \text{cost}(\alpha_t^{\star})$;
17     $t \leftarrow t + 1$;
18 **return** *Final output derived from* $\mathcal{H}_t$

| Method | Budget ($) | Total Cost ($) | Accuracy (%) |
|---|---|---|---|
| With Modules | | | |
| + No Budget Aware | | 0.1156 | 35.80 |
| + Budget-Aware Prompting | 0.2 | 0.1246 | 36.41 |
| + Best-of-$N$ Sampling | | 0.1698 | 35.80 |
| + Iterative Verification | | 0.1928 | 36.41 |
| AGENT* | | 0.1660 | **38.89** |
| With Modules | | | |
| + No Budget Aware | | 0.1409 | 37.65 |
| + Budget-Aware Prompting | 0.3 | 0.1592 | 41.97 |
| + Best-of-$N$ Sampling | | 0.2583 | 37.65 |
| + Iterative Verification | | 0.2903 | 42.59 |
| AGENT* | | 0.2233 | **44.44** |
| With Modules | | | |
| + No Budget Aware | | 0.1507 | 38.27 |
| + Budget-Aware Prompting | 0.4 | 0.1727 | 43.20 |
| + Best-of-$N$ Sampling | | 0.3464 | 43.20 |
| + Iterative Verification | | 0.3885 | 43.20 |
| AGENT* | | 0.2771 | **46.91** |
| With Modules | | | |
| + No Budget Aware | | 0.1587 | 38.27 |
| + Budget-Aware Prompting | 0.5 | 0.1771 | 43.20 |
| + Best-of-$N$ Sampling | | 0.3825 | 44.44 |
| + Iterative Verification | | 0.4803 | 43.20 |
| AGENT* | | 0.3090 | **48.15** |

Table 6: Performance comparison among different test-time scaling methods under budget constraints on GAIA with Claude.

