# OpenReview forum: "AGENT*: Optimizing Test-Time Compute for Multi-Agent Systems with Modularized Collaboration"
_ICLR.cc/2026/Conference — Submitted to ICLR 2026_

### Official Review · Reviewer_cyXt · 2025-10-15

**Soundness:** 3
**Presentation:** 3
**Contribution:** 3
**Rating:** 4
**Confidence:** 3

**Summary:**

The paper introduces AGENT*, a framework that optimizes test-time computation in multi-agent systems under fixed compute budgets. It proposes collaboration modules that capture reusable coordination strategies and a dual-level planning architecture combining short-term decision making with long-term budget estimation. Experiments on GAIA and BrowseComp-Plus benchmarks using Claude and Qwen3 models show improved success rates and more efficient compute utilization compared to orchestration and budget-aware baselines.

**Strengths:**

1. Clearly defines an important and emerging problem on managing inference-time compute for multi-agent systems.
2. The dual-level planning structure provides an intuitive way to coordinate agents while maintaining compute constraints.
3. The experiments are systematic and demonstrate consistent performance gains across several models and budget settings.

**Weaknesses:**

1. Limited empirical diversity Experiments are conducted only on GAIA and BrowseComp-Plus, both text-based web-agent benchmarks. The method’s generality for other multi-agent settings (e.g., embodied collaboration, code generation, or reasoning-only domains) remains unverified, making it unclear whether AGENT*’s benefits generalize beyond these specific tasks.
2. Evaluation metrics and significance. The reported gains (roughly 2–4% absolute accuracy improvement) are modest and lack statistical validation such as confidence intervals or significance testing. It is difficult to judge whether these differences are meaningful given the stochasticity of LLM outputs and sampling variability.
3. Unclear cost realism and reproducibility. The “budget” is defined in monetary units ($0.2–0.5) without clear mapping to real compute (tokens, latency, or FLOPs). This abstraction complicates interpreting efficiency claims and reproducing the budgeted setting on other models or infrastructures.

**Questions:**

1. How exactly is the “budget” quantified during experiments? Is it based on token usage, latency, or monetary cost estimates from API calls? A clearer and reproducible definition would help assess whether the efficiency improvements are practical and transferable to other model families.
2. Could the authors provide an ablation isolating the effects of the short-term and long-term planning components? It would help determine whether both levels are essential, or if most of the gains come from one part of the dual-level design.

---

> ### Author Response · Authors · 2025-11-22
> **Response to Reviewer cyXt (1/2)**
>
> We sincerely thank the reviewer for their encouraging feedback, particularly their recognition of our clear problem formulation, the intuitiveness of our dual-level planning architecture, and the systematic experiments demonstrating consistent gains across models and budget regimes. Below, we address your concerns and clarify key points.
>
> > **Weakness 1**: Limited empirical diversity Experiments are conducted only on GAIA and BrowseComp-Plus, both text-based web-agent benchmarks. The method’s generality for other multi-agent settings (e.g., embodied collaboration, code generation, or reasoning-only domains) remains unverified, making it unclear whether AGENT*’s benefits generalize beyond these specific tasks.
>
> We selected GAIA and BrowseComp-Plus because our goal is to evaluate AGENT* on tasks that **truly require multi-agent collaboration**. These two benchmarks are the primary existing datasets that involve heterogeneous agents with distinct specialties, making them well-suited for assessing our framework. In particular, they require a wide range of capabilities, including search, browsing, document retrieval, long-document reading, and multimodal reasoning, each necessitating specialized agents equipped with dedicated tools.
>
> Although there are other benchmarks in domains such as web navigation and coding (e.g., WebArena, WebShop, SWE-bench), these tasks are typically solvable by single-agent systems with tool APIs. Since they do not meaningfully benefit from or require coordinated multi-agent interactions, they are not appropriate for evaluating the core contributions of AGENT*.
>
> > **Weakness 2**: Evaluation metrics and significance. The reported gains (roughly 2–4% absolute accuracy improvement) are modest and lack statistical validation such as confidence intervals or significance testing. It is difficult to judge whether these differences are meaningful given the stochasticity of LLM outputs and sampling variability.
>
> The strongest external baseline in our evaluation is ReAct w/ Iterative Verification and compared to this baseline, our method yields improvements that are substantially larger than +2–4% in nearly all settings. For example, on GAIA with Claude models, AGENT* exceeds ReAct w/ Iterative Verification by +3–11 points across budgets, and similarly achieves +3–5 point gains on GAIA with Qwen3-32B as well as +4–10 point gains on BrowseComp-Plus with Qwen3-32B. These margins demonstrate that our improvements over the strongest baseline are **meaningfully large and far beyond what could be attributed to the stochasticity of LLM outputs or sampling variability**.
>
> Importantly, both GAIA and BrowseComp-Plus are highly challenging benchmarks where prior studies have shown that achieving even +1–2 point absolute gains on these datasets is difficult and often reflects meaningful progress in complex reasoning ability. **The consistent improvements we observe across all budgets, models, and datasets indicate that our method provides a substantive and reliable performance advancement, not an artifact of randomness**.
>
> > **Weakness 3 & Question 1**: Unclear cost realism and reproducibility. The “budget” is defined in monetary units ($0.2–0.5) without clear mapping to real compute (tokens, latency, or FLOPs). This abstraction complicates interpreting efficiency claims and reproducing the budgeted setting on other models or infrastructures. How exactly is the “budget” quantified during experiments? Is it based on token usage, latency, or monetary cost estimates from API calls? A clearer and reproducible definition would help assess whether the efficiency improvements are practical and transferable to other model families.
>
> We apologize for the confusion. **Our “budget” is computed directly from the model’s token pricing**. Specifically, we use the U.S. dollar cost defined by the input and output token rates published in the AWS Bedrock pricing page (https://aws.amazon.com/bedrock/pricing/):
>
> anthropic/claude-3-5-haiku-latest: input \$0.0000008/token, output \$0.000004/token \
> anthropic/claude-3-7-sonnet-latest: input \$0.000003/token, output \$0.000015/token \
> qwen3-32b: input \$0.0000007/token, output \$0.0000028/token
>
> Thus, every action’s cost is computed exactly from its input/output token usage, and the total budget corresponds to a fixed token budget. This definition is inherently model-agnostic: any model family with known token pricing can be plugged in, making it practical and transferable. We have clarified this in lines 308–309 and in Appendix B.

---

> > ### Comment · Reviewer_cyXt · 2025-11-23
> >
> > I thank the authors for their detailed response.
> >
> > The rebuttal effectively clarifies that the budget is strictly derived from token usage, resolving my reproducibility concerns, and I acknowledge that Table 2  already addresses the requested ablation study. While I maintain that the empirical scope remains somewhat limited to web-based tasks.

---

> > > ### Author Response · Authors · 2025-11-26
> > > **Response to Reviewer cyXt**
> > >
> > > Thanks for your response and for adjusting the scores. To address your concern about the limited domain of our evaluation benchmarks, **we conducted an additional experiment on BigCodeBench to validate our findings beyond GAIA and BrowseComp-Plus**. The full experimental results are provided in the official comment above. **In summary, we observed that our method exhibits the same improvement trends on this coding benchmark, reinforcing that our method generalizes well beyond web- and search-based tasks.** We hope this addresses your concerns about the empirical scope limitation.

---

> ### Author Response · Authors · 2025-11-22
> **Response to Reviewer cyXt (2/2)**
>
> > **Question 2**: Could the authors provide an ablation isolating the effects of the short-term and long-term planning components? It would help determine whether both levels are essential, or if most of the gains come from one part of the dual-level design.
>
> | Method            | Acc@0.2 | Acc@0.5 |
> | ----------------- | ------- | ------- |
> | Only Short-term   | 36.41   | 44.44   |
> | Only Long-term    | 37.65   | 46.91   |
> | Dual-level (Both) | 38.89   | 48.15   |
>
> We have added comprehensive ablation studies in Section 4.4 on dual-level planning, where we isolate each component and report Accuracy under budgets of \$0.2 and \$0.5. Short-term planning improves over vanilla ReAct by guiding better local decisions and promoting effective module usage. Long-term planning performs slightly better, especially at higher budgets because it anticipates future computation and avoids early suboptimal choices. Combining both yields the best results, showing that integrating local and global planning leads to the most budget-efficient multi-agent coordination.

---

### Official Review · Reviewer_XHcw · 2025-10-30

**Soundness:** 3
**Presentation:** 3
**Contribution:** 3
**Rating:** 6
**Confidence:** 3

**Summary:**

This paper proposes Agent*, a framework for budget-aware multi-agent collaboration. The framework introduces reusable collaboration modules that the orchestrator can invoke directly, and selects actions using a dual-level planning architecture that balances short-term task completion and long-term speculation under budget constraints. Experiments on GAIA and BrowseComp-Plus show that Agent* achieves higher performance than baselines and utilizes the budget more effectively.

**Strengths:**

1.	The paper is clearly written and addresses an important and practical issue: multi-agent reasoning under constrained budgets.
2.	The design of Agent* is intuitive and well-structured. It incorporates commonly used components such as reusable skills and self-consistency sampling, organizing them in a coherent and elegant manner.
3.	Experimental results demonstrate that both the collaboration modules and budget-aware planning contribute effectively, and that Agent* outperforms standard test-time scaling methods with comparable costs.

**Weaknesses:**

1. The related work section should include papers that explore budget constraints in agentic reasoning, e.g., [1] [2].

[1] Zheng, Yuanhang, et al. "Budget-Constrained Tool Learning with Planning." Findings of the Association for Computational Linguistics ACL 2024. 2024.

[2] Qiu, Rennai, et al. "Co-Saving: Resource Aware Multi-Agent Collaboration for Software Development." arXiv preprint arXiv:2505.21898 (2025).

2. The current baselines function more as ablation studies within the proposed framework. Additional comparisons with other multi-agent collaboration and orchestration frameworks would strengthen the validation of Agent*’s effectiveness.

**Questions:**

1.	Why is normalization applied to the budget feasibility score? Does this favor trajectories with lower costs and cause the average cost to remain consistently below the budget?
2.	Could you provide an analysis of how Agent* allocates resources? Does it explicitly perform test-time scaling for particularly difficult tasks?

---

> ### Author Response · Authors · 2025-11-22
> **Response to Reviewer XHcw**
>
> We sincerely appreciate the reviewer’s positive assessment of our work, especially the recognition of our clear presentation, intuitive system design, and strong empirical results. Below, we address your concerns and clarify key points.
>
> > **Weakness 1**: The related work section should include papers that explore budget constraints in agentic reasoning, e.g., [1] [2].
>
> We have updated the related work section to include papers related to budget constraint settings, including  [1,2] in lines 442-448. While these works have explored budget-constrained settings, they either focus on a single LLM setting or target system-level efficiency rather than scaling test-time computation to improve performance. In contrast, our work focuses on budget-optimal compute allocation in a multi-agent setting, which is not addressed by these prior approaches
>
> [1] Zheng, Yuanhang, et al. "Budget-Constrained Tool Learning with Planning." Findings of the Association for Computational Linguistics ACL 2024. 2024. \
> [2] Qiu, Rennai, et al. "Co-Saving: Resource Aware Multi-Agent Collaboration for Software Development." arXiv preprint arXiv:2505.21898 (2025).
>
> > **Weakness 2**: The current baselines function more as ablation studies within the proposed framework. Additional comparisons with other multi-agent collaboration and orchestration frameworks would strengthen the validation of Agent*’s effectiveness.
>
> We have additionally added ADAS [1] and AFlow [2] as our baselines to compare the performance with the recent agent workflow optimization works. Fixed agent workflow baselines underperform on both GAIA and BrowseComp-Plus because they cannot adapt agent execution dynamically during inference. These benchmarks contain questions that require highly diverse workflows, causing the optimized workflow in these methods to collapse into a simple, generic pipeline, which makes it difficult for a single, static workflow to address these tasks effectively. Moreover, since these approaches have no mechanism for dynamically allocating compute, their performance remains largely unchanged even as the available budget increases.
>
> [1] Hu et al, Automated design of agentic systems. ICLR 2025 \
> [2] Zhang et al, Aflow: Automating agentic workflow generation. ICLR 2025
>
> > **Question 1**: Why is normalization applied to the budget feasibility score? Does this favor trajectories with lower costs and cause the average cost to remain consistently below the budget?
>
> The normalization in the budget feasibility score is applied to keep the long-term score $h$ on the same scale (0-1) as the short-term gain $g$ when computing the overall utility $f = g+h$. Normalization ensures that both components contribute comparably to the action selection.
>
> > **Question 2**: Could you provide an analysis of how Agent* allocates resources? Does it explicitly perform test-time scaling for particularly difficult tasks?
>
> AGENT* adaptively allocates resources at the subtask level, selecting stronger collaboration modules for difficult or high-impact subtasks and lighter agents for simpler ones. Here is an actual trajectory from our experiment for the question:
>
> Question: What was the volume in m³ of the fish bag calculated in the University of Leicester paper “Can Hiccup Supply Enough Fish to Maintain a Dragon’s Diet?”
>
> Trajectory:
> ensemble_search_agent(“Find the University of Leicester paper and the exact fish-bag volume…”)
>  → browser_agent(“Please browse https://journals.le.ac.uk/… to verify the value”)
>
> In this example, the system identifies the first subtask (locating the full paper and extracting the precise volume) as more demanding and assigns the more powerful ensemble_search_agent module. Once the URL is identified, the subsequent subtask (verifying the value from the source) is straightforward, and the planner assigns browser_agent. This demonstrates AGENT*’s ability to allocate compute based on subtask importance and difficulty, rather than applying uniform scaling at the task level, effectively enabling fine-grained test-time scaling.

---

> > ### Comment · Reviewer_XHcw · 2025-11-26
> >
> > Thank the authors for the rebuttal. It addresses most of my concerns, and it would be great to see the comparison with methods like GPTSwarm, as suggested by reviewer jpH3.

---

### Official Review · Reviewer_9zKN · 2025-11-01

**Soundness:** 3
**Presentation:** 2
**Contribution:** 2
**Rating:** 4
**Confidence:** 3

**Summary:**

This paper introduces AGENT*, a framework for optimizing test-time compute allocation in multi-agent systems (MAS) under fixed inference-time budgets. Experiments on GAIA and BrowseComp-Plus benchmarks comparing AGENT* with several baselines (no modules, with modules, budget-aware prompting, and test-time scaling methods like best-of-N and iterative verification). The paper claims that AGENT* consistently outperforms baselines in both accuracy and budget utilization, demonstrating more efficient test-time scaling for collaborative agents.

**Strengths:**

1. This paper tries to address an important and timely problem, the efficient inference-time scaling for multi-agent systems.

**Weaknesses:**

1. The numerical improvements are within standard deviation of baselines. Accuracy improvements (≈ +2–3 %) are modest and not statistically validated
2. Only GAIA and BrowseComp-Plus are used. It would be better to evaluate on more benchmarks to test the efficacy of the method on diverse domains.
3. The paper lacks experiments isolating each component (dual-level planner, self-play reflection, cost estimation). Without these, it’s impossible to know what truly drives performance.
4. The related-work section misses recent agentic workflow optimization and inference-time orchestration papers
5. The “budget” concept is abstracted into token cost, but real inference latency and system cost are not reported.
6. Figures and captions are poorly formatted. Many fonts are too small. quantitative data tables are cluttered;

**Questions:**

1. How sensitive are results to the number of self-play rounds or to the number of collaboration modules derived?
2. Would AGENT* still outperform when all systems are trained with the same cost-budgeted objective (e.g., Muennighoff et al., 2025 “S1”)?
3. How often are collaboration modules reused across tasks, and do they transfer?

---

> ### Author Response · Authors · 2025-11-22
> **Response to Reviewer 9zKN (1/2)**
>
> We appreciate the reviewer for highlighting the importance of our problem setup and solutions. Below, we address your concerns and clarify key points.
>
> > **Weakness 1**: The numerical improvements are within standard deviation of baselines. Accuracy improvements (≈ +2–3 %) are modest and not statistically validated
>
> The strongest external baseline in our evaluation is ReAct w/ Iterative Verification and compared to this baseline, our method yields improvements that are **substantially larger than +2–3% in nearly all settings**. For example, on GAIA with Claude models, AGENT* exceeds ReAct w/ Iterative Verification by +3–11 points across budgets, and similarly achieves +3–5 point gains on GAIA with Qwen3-32B as well as +4–10 point gains on BrowseComp-Plus with Qwen3-32B. These margins demonstrate that our improvements over the strongest baseline are meaningfully large and not within typical noise levels.
> Importantly, both GAIA and BrowseComp-Plus are highly challenging benchmarks where prior studies have shown that achieving even +1–2 point absolute gains on these datasets is difficult and often reflects meaningful progress in complex reasoning ability.
>
> > **Weakness 2**: Only GAIA and BrowseComp-Plus are used. It would be better to evaluate on more benchmarks to test the efficacy of the method on diverse domains.
>
> We selected GAIA and BrowseComp-Plus because our goal is to evaluate AGENT* on **tasks that truly require multi-agent collaboration**. These two benchmarks are the primary existing datasets that involve heterogeneous agents with distinct specialties, making them well-suited for assessing our framework. In particular, they require a wide range of capabilities including search, browsing, document retrieval, long-document reading, and multimodal reasoning, each necessitating specialized agents equipped with dedicated tools.
>
> Although there are other benchmarks in domains such as web navigation and coding (e.g., WebArena, WebShop, SWE-bench), these tasks are typically solvable by single-agent systems with tool APIs. Since they do not meaningfully benefit from or require coordinated multi-agent interactions, they are not appropriate for evaluating the core contributions of AGENT*.
>
> > **Weakness 3**: The paper lacks experiments isolating each component (dual-level planner, self-play reflection, cost estimation). Without these, it’s impossible to know what truly drives performance.
>
> | Method            | Acc@0.2 | Acc@0.5 |
> | ----------------- | ------- | ------- |
> | Only Short-term   | 36.41   | 44.44   |
> | Only Long-term    | 37.65   | 46.91   |
> | Dual-level (Both) | 38.89   | 48.15   |
>
> We have added a comprehensive ablation study in Section 4.4 on dual-level planning, where we isolate each component and report Accuracy under budgets of $0.2 and $0.5. Short-term planning improves over vanilla ReAct by guiding better local decisions and promoting effective module usage. Long-term planning performs slightly better, especially at higher budgets because it anticipates future computation and avoids early suboptimal choices. Combining both yields the best results, showing that integrating local and global planning leads to the most budget-efficient multi-agent coordination.
>
> > **Weakness 4**: The related-work section misses recent agentic workflow optimization and inference-time orchestration papers
>
> We would like to clarify that our related work section already includes recent agent workflow optimization methods such as ADAS [1] and AFlow [2]. In addition, we have now incorporated even more recent approaches, including MaaS [3] and FlowReasoner [4]. We have also added ADAS and AFlow as baselines in Table 1 to ensure a comprehensive comparison with the latest advances in agent workflow optimization.
>
> [1] Hu et al, Automated design of agentic systems. ICLR 2025 \
> [2] Zhang et al, Aflow: Automating agentic workflow generation. ICLR 2025 \
> [3] Zhang et al, Multi-agent architecture search via agentic supernet. ICML 2025 \
> [4] Gao et al, Flowreasoner: Reinforcing query-level meta-agents. Arxiv

---

> ### Author Response · Authors · 2025-11-22
> **Response to Reviewer 9zKN (2/2)**
>
> > **Weakness 5**: The “budget” concept is abstracted into token cost, but real inference latency and system cost are not reported.
>
> We use token cost as the “budget” because it provides a **consistent and reproducible measure of test-time compute**. In contrast, inference latency is highly variable since it is affected by multiple confounding engineering factors like network/API delays, making it unreliable as a controlled metric for algorithmic evaluation. Also, token cost **reflects real deployment, where users face a hard monetary compute budget** (directly tied to tokens). Importantly, AGENT* is not restricted to this choice, the planner only requires a scalar cost, so the same framework works with latency or system-level cost without any algorithmic changes. Although we do not report latency, it is generally positively correlated with token usage, so improvements in token-budget efficiency also translate to reduced compute overhead.
>
> > **Weakness 6**: Figures and captions are poorly formatted. Many fonts are too small. quantitative data tables are cluttered;
>
> We revised Table 1 to make the comparison more clear by reorganizing our baselines into three categories: (1) fixed agent workflow, (2) ReAct with test-time scaling, and (3) ReAct with collaboration modules. We also introduced a new metric, Acc@B (lines 316–320), which denotes the accuracy achieved under a given budget B. Lastly, we have made the figure fonts bigger for better readability (Figure 2).
>
> > **Question 1**: How sensitive are results to the number of self-play rounds or to the number of collaboration modules derived?
>
> Each self-play round is designed to extract one collaboration module using the self-play reflection prompt in Figure 5. In practice, we find the results to be insensitive to the exact number of rounds because the space of meaningful collaboration patterns is limited by the capabilities and roles of the available agents. For example, in the BrowseComp-Plus experiments, the final round produced a duplicate module (ensemble_interactive_search_then_critic), indicating that additional rounds do not yield new coordination structures given the three available agents. This suggests that self-play reflection naturally converges once all useful collaboration patterns have been discovered.
>
> > **Question 2**: Would AGENT* still outperform when all systems are trained with the same cost-budgeted objective (e.g., Muennighoff et al., 2025 “S1”)?
>
> Even if a system was optimized under the same cost-budgeted objective, AGENT* would still offer advantages because S1 and similar approaches do not address the core challenge we study. S1 enforces a fixed per-sample budget but provides no mechanism for deciding how to schedule or coordinate multiple agents under a shared budget. Multi-agent systems **require dynamic decisions about which agent or collaborative workflow to invoke and when, something S1 does not optimize**.
>
> Moreover, a cost-budgeted objective alone cannot adapt to different agentic systems or environments. Different multi-agent setups involve distinct collaboration structures and cost profiles, and **simply applying a fixed budget constraint (as in S1) does not give the system the ability to adjust its compute allocation accordingly**. In contrast, AGENT* performs inference-time planning, allowing it to adapt dynamically to the available agents, their costs, and the remaining budget.
>
> Thus, even under the same cost-budgeted objective, AGENT* would retain its advantage because multi-agent compute scheduling requires dynamic, environment-aware reasoning, not just a fixed budget constraint.
>
> > **Question 3**: How often are collaboration modules reused across tasks, and do they transfer?
>
> To assess the transferability of collaboration modules induced from the 30 validation questions, we examined how often they are reused in the 162 test questions in GAIA benchmark for the Claude experiments. We find that 61.73% of test questions (100 out of 162) invoked at least one collaboration module, demonstrating that the extracted modules generalize beyond the validation set and are reused on a substantial portion of unseen tasks. Moreover, this reuse correlates with improved performance: under the $0.5 budget, AGENT* achieves 48.1% accuracy, compared to 36.4% for the No Module, No Budget Aware baseline, showing that AGENT* effectively leverages these transferable modules to achieve significant gains.

---

> ### Author Response · Authors · 2025-11-26
> **Response to Reviewer 9zKN**
>
> To better address your concern about the limited domain of our evaluation benchmarks, **we conducted an additional experiment on BigCodeBench to validate our findings beyond GAIA and BrowseComp-Plus**. The full experimental results are provided in the official comment above. In summary, we observed that **our method exhibits the same improvement trends on this coding benchmark, reinforcing that our method generalizes well beyond web- and search-based tasks**. We hope this addresses your concerns about the efficacy of our method in other domains.
>
> As the discussion phase is coming to an end, we would be grateful to hear whether our responses address your comments. Thank you again for your valuable review.

---

### Official Review · Reviewer_jpH3 · 2025-11-01

**Soundness:** 2
**Presentation:** 2
**Contribution:** 3
**Rating:** 4
**Confidence:** 5

**Summary:**

This paper investigates the problem of optimizing **test-time scaling (TTS)** strategies within **multi-agent systems (MAS)**. The authors employ **self-play** on a validation set to automatically discover potential **collaboration modules** and their corresponding computational costs, thereby expanding the search space beyond that of a single agent. During inference, the system leverages this expanded space to solve complex tasks for each query in a **budget-aware**, stepwise manner, following an **orchestrator–worker** collaboration paradigm to adaptively select appropriate agents or modules.

**Strengths:**

1. **Novel research problem.** This paper extends the optimization of test-time scaling to multi-agent systems under the orchestrator–worker paradigm, exploring how to maximize overall system performance and effectively utilize individual agent capabilities under a fixed computational budget. The problem setting demonstrates strong research value.

2. Experimental results show that the proposed collaboration modules and dual-level planning mechanisms effectively enhance the performance utilization of multi-agent systems under budget constraints, validating the practical potential of the proposed approach.

**Weaknesses:**

1. **Insufficient baseline design.** The paper lacks well-designed comparative baselines, making it difficult to demonstrate the state-of-the-art effectiveness of the proposed method convincingly.
   (1) In terms of automated agent collaboration optimization, the paper does not compare against recent works that explicitly optimize multi-agent coordination—such as GPTSwarm [1], G-Desiger [2], AFlow [3], MAS-GPT, and TTS-related agent scaling [4], [5], [6]—whose underlying philosophy is closely related to the proposed budget-aware orchestration mechanism.
   (2) Regarding the TTS setting, the authors only compare one way of utilization of two TTS strategies (best-of-N and iterative verification) which include the TTS-related modules in the agent space. Still, they omit other simple combination strategies of TTS and agents, e.g., applying TTS directly to the original single-agent in each subtask (for instance, dynamically assigning budget to each agent under TTS strategies such as *Best-of-N*).

[1] Zhuge, Mingchen, et al. *“GPTSwarm: Language Agents as Optimizable Graphs.”* *Forty-first International Conference on Machine Learning (ICML)*, 2024.
[2] Zhang, Guibin, et al. "G-designer: Architecting multi-agent communication topologies via graph neural networks." arXiv preprint arXiv:2410.11782 (2024).
[3] Zhang, Jiayi, et al. "Aflow: Automating agentic workflow generation." arXiv preprint arXiv:2410.10762 (2024).
[4] Ye, Rui, et al. "MAS-GPT: Training LLMs to build LLM-based multi-agent systems." arXiv preprint arXiv:2503.03686 (2025).
[5] Jin, Can, et al. "Two heads are better than one: Test-time scaling of multi-agent collaborative reasoning." arXiv preprint arXiv:2504.09772 (2025).
[6] Qian, Chen, et al. "Scaling large language model-based multi-agent collaboration." arXiv preprint arXiv:2406.07155 (2024).


2. **Misalignment between research goals and methodology.** Although the paper claims to aim for “performance maximization,” the proposed method lacks an explicit mechanism for performance optimization. For example, the short-term planning component relies solely on self-consistency scoring and uses a single LLM to generate execution strategies. It remains unclear whether this LLM can adaptively select the optimal collaboration pattern for performance across diverse tasks.

3. **Insufficient methodological details.**
   (1) Although the authors mention that their policy generation follows the *ReAct* framework, they do not clarify how candidate actions are concretely generated under the proposed setting (e.g., prompt templates, sampling strategies).
   (2) The description of the *self-play* process omits key implementation details, such as which TTS techniques are used, how experience is store, updated, and summarized, and how the number of sampled modules is determined.

4. **Lack of discussion on model scaling effects.** The paper does not address performance differences between small and large models under equivalent computational budgets—an essential topic in TTS research. Prior work [7] has shown that smaller models can sometimes outperform large models given equivalent compute resources.

[7] Snell, Charlie, et al. *“Scaling LLM Test-Time Compute Optimally Can Be More Effective than Scaling Model Parameters.”* *arXiv preprint arXiv:2408.03314*, 2024.

5. **Questionable use of a validation set.** The paper aims to address query-level inference scenarios, yet it relies on a validation set to discover collaboration modules. This assumption is unrealistic, as no such validation set would be available in real-world inference settings to support experience-based module selection.

6. **Spelling and grammatical issues.** Several minor language problems reduce readability. For instance, in Section 2, “each initizlied with a large language model (LLM)” should be corrected to “initialized.”

7. **Incomplete citations.** Some key references are missing. For example, line 189 (“A* algorithm”) and line 46 (“orchestrator–worker”) should be accompanied by appropriate citations.

8. **Ambiguity in terminology.** The term *“collaboration module”* is potentially misleading, as it suggests a method component rather than an *agent group*. The authors are advised to adopt a clearer term to better reflect its intended meaning.

**Questions:**

1. The paper states that “we conduct five rounds of self-play reflection for each benchmark, with each round producing one candidate module.” Would increasing the number of reflection rounds further improve overall performance?

2. Which specific **test-time scaling** techniques are employed in the paper? Does the approach incorporate strategies such as *Best-of-N*, *Self-Refinement*, or other advanced TTS strategies?

3. How does the proposed algorithm ensure performance maximization in its design? Are there explicit optimization objectives or adaptive mechanisms?

4. What is the specific role of the symbol $\kappa$ in the formula at line 171?

5. Is the task step number $H$ fixed or dynamically determined? If it is dynamic, how does this affect the overall budget constraint?

6. Can the proposed method be extended to other agent collaboration ways beyond orchestrator–worker multi-agent system configurations, like debate-based collaboration?

---

> ### Author Response · Authors · 2025-11-23
> **Response to Reviewer jpH3 (1/3)**
>
> We appreciate the reviewer’s recognition of our novel problem setting and their positive feedback on the effectiveness of our collaboration modules and dual-level planning. Below, we address your concerns and clarify key points.
>
> > **Weakness 1**: Insufficient baseline design. The paper lacks well-designed comparative baselines, making it difficult to demonstrate the state-of-the-art effectiveness of the proposed method convincingly. (1) In terms of automated agent collaboration optimization, the paper does not compare against recent works that explicitly optimize multi-agent coordination.  (2) Regarding the TTS setting, the authors only compare one way of utilization of two TTS strategies (best-of-N and iterative verification) which include the TTS-related modules in the agent space. Still, they omit other simple combination strategies of TTS and agents
>
> **(1) Missing recent multi-agent collaboration optimization baselines**
>
> To address your concerns on the lack of baselines, we have additionally added ADAS [1] and AFlow [2] as our baselines to compare the performance with the recent agent workflow optimization works. Fixed agent workflow baselines underperform on both GAIA and BrowseComp-Plus because they cannot adapt agent execution dynamically during inference. These benchmarks contain questions that require highly diverse workflows, causing the optimized workflow in these methods to collapse into a simple, generic pipeline, which makes it difficult for a single, static workflow to address these tasks effectively. Moreover, since these approaches have no mechanism for dynamically allocating compute, their performance remains largely unchanged even as the available budget increases.
>
> Other systems such as GPTSwarm [3], G-Designer [4], MAS-GPT [5], and MaAS [6] typically require model or system training / fine-tuning over environment trajectories (e.g., policy-gradient training over communication graphs, or architecture supernet training). Integrating them as baselines on GAIA and BrowseComp-Plus would therefore require training new, environment-specific policies, which is orthogonal to our focus on training-free inference-time planning for a fixed pool of agents under a strict budget.
>
> Importantly, many of these systems optimize a single architecture/workflow per task and then execute it, rather than performing test-time, budget-aware, adaptive planning across multiple reusable subworkflows, which is the core contribution of AGENT*.
>
> **(2) Other ways of combining TTS strategies and agents**
>
> We believe that Best-of-N and Iterative Verification are the most widely used and adopted test-time scaling methods in multi-agent systems. As Table 1 shows, these TTS baselines consume more compute but yield only small and inconsistent gains, and their performance quickly plateaus as the budget grows. We believe that the variations of these TTS baselines will also show similar results since these TTS methods lack a mechanism to reason over which agents or workflows should receive extra compute and when, under a shared global budget.
>
> [1] Hu et al, Automated design of agentic systems. ICLR 2025 \
> [2] Zhang et al, Aflow: Automating agentic workflow generation. ICLR 2025 \
> [3] Zhuge, Mingchen, et al. “GPTSwarm: Language Agents as Optimizable Graphs.ICML 2024 \
> [4] Zhang, Guibin, et al. G-designer: Architecting multi-agent communication topologies via graph neural networks. arXiv \
> [5] Ye, Rui, et al. MAS-GPT: Training LLMs to build LLM-based multi-agent systems. arXiv \
> [6] Zhang et al, Multi-agent architecture search via agentic supernet. ICML 2025
>
> > **Weakness 2**: Misalignment between research goals and methodology. Although the paper claims to aim for “performance maximization,” the proposed method lacks an explicit mechanism for performance optimization.
>
> We believe that self-consistency is an effective choice for short-term planning because it provides a reliable, model-internal estimate of action quality: when multiple stochastic samples independently select the same agent or collaboration module, the model is signaling strong confidence that this workflow is appropriate for the current subtask. This principle underlies many successful test-time scaling methods (e.g., Best-of-N, CoT-SC), and here it serves as a lightweight but robust signal for identifying locally optimal collaboration patterns using test-time compute without any training involved.
>
> Also, AGENT* does not rely on self-consistency alone. The long-term planner evaluates whether a candidate action leaves enough remaining budget for the rest of the trajectory. This joint mechanism allows AGENT* to dynamically select the most effective collaboration pattern for each subtask and each budget state, rather than committing to a fixed workflow or relying solely on local heuristics, which explains its consistent gains across diverse tasks and budgets.

---

> ### Author Response · Authors · 2025-11-23
> **Response to Reviewer jpH3 (2/3)**
>
> > **Weakness 3**: Insufficient methodological details. (1) Although the authors mention that their policy generation follows the ReAct framework, they do not clarify how candidate actions are concretely generated under the proposed setting (e.g., prompt templates, sampling strategies). (2) The description of the self-play process omits key implementation details, such as which TTS techniques are used, how experience is store, updated, and summarized, and how the number of sampled modules is determined.
>
> For candidate action generation, we clarify that the orchestrator is prompted to output agent calls using a tool-call style format (e.g., < tool_call >...< /tool_call >), mirroring modern LLM tool-use interfaces. Our prompting templates are adapted from smolagents library (https://github.com/huggingface/smolagents), ensuring that agent/module invocation follows a standard, reproducible ReAct-style structure. For the self-play reflection process, we already provided the exact prompt used to induce collaboration modules in Figure 3. This prompt shows how trajectories are summarized and how recurring patterns are transformed into reusable modules.
>
> > **Weakness 4**: Lack of discussion on model scaling effects. The paper does not address performance differences between small and large models under equivalent computational budgets—an essential topic in TTS research. Prior work [7] has shown that smaller models can sometimes outperform large models given equivalent compute resources.
>
> We would like to clarify that our goal is not to compare the raw performance of small vs. large models under identical compute budgets like in [1]. Instead, our objective is to evaluate how well the proposed dual-level planner adapts to diverse budget constraints when coordinating heterogeneous agents.
>
> To ensure that our framework is generalizable across different model capacities, we intentionally include both Claude-based agents and Qwen3-32B, which differ substantially in capability and token cost. This setup tests whether AGENT* can effectively allocate compute across agents of varying strengths, rather than testing parameter-scaling laws. Our method is orthogonal to model scaling: once token costs are known, the planner can operate over any mixture of small and large agents.
>
> [1] Snell, Charlie, et al. “Scaling LLM Test-Time Compute Optimally Can Be More Effective than Scaling Model Parameters.” arXiv preprint arXiv:2408.03314, 2024.
>
> > **Weakness 5**: Questionable use of a validation set. The paper aims to address query-level inference scenarios, yet it relies on a validation set to discover collaboration modules. This assumption is unrealistic, as no such validation set would be available in real-world inference settings to support experience-based module selection.
>
> We believe that as an entirely training-free framework, using a small quantity of validation set is a feasible setting. The only purpose of the 30-example validation set is to provide a small amount of warm-start experience for (i) estimating average agent/module costs and (ii) inducing reusable collaboration modules through self-play reflection. This is a very feasible and realistic assumption, as most real-world deployments naturally accumulate a small set of representative user queries or internal test questions before full rollout.
>
> > **Weakness 6**: Spelling and grammatical issues. Several minor language problems reduce readability. For instance, in Section 2, “each initizlied with a large language model (LLM)” should be corrected to “initialized.”
>
> We have reviewed the manuscript and corrected the noted spelling and grammatical errors, including the example provided. The updated version includes a thorough pass to ensure clarity and readability throughout.
>
> > **Weakness 7**: Incomplete citations. Some key references are missing. For example, line 189 (“A* algorithm”) and line 46 (“orchestrator–worker”) should be accompanied by appropriate citations.
>
> We have added the missing citations mentioned by the reviewer, including references for A*, orchestrator–worker systems in the revised manuscript to ensure all claims and terminology are properly attributed.
>
> > **Weakness 8**:  Ambiguity in terminology. The term “collaboration module” is potentially misleading, as it suggests a method component rather than an agent group. The authors are advised to adopt a clearer term to better reflect its intended meaning.
>
> We use the term “collaboration modules” to emphasize that these components modulate multi-agent workflows: each module encapsulates a coordinated interaction pattern among agents, not a standalone method component.

---

> ### Author Response · Authors · 2025-11-23
> **Response to Reviewer jpH3 (3/3)**
>
> > **Question 1**: The paper states that “we conduct five rounds of self-play reflection for each benchmark, with each round producing one candidate module.” Would increasing the number of reflection rounds further improve overall performance?
>
> Each self-play round is designed to extract one collaboration module using the self-play reflection prompt in Figure 5. In practice, we find the results to be insensitive to the exact number of rounds because the space of meaningful collaboration patterns is limited by the capabilities and roles of the available agents. For example, in the BrowseComp-Plus experiments, the final round produced a duplicate module (ensemble_interactive_search_then_critic), indicating that additional rounds do not yield new coordination structures given the three available agents. This suggests that self-play reflection naturally converges once all useful collaboration patterns have been discovered.
>
> > **Question 2**: Which specific test-time scaling techniques are employed in the paper? Does the approach incorporate strategies such as Best-of-N, Self-Refinement, or other advanced TTS strategies?
>
> In AGENT*, using collaboration modules already serves as test-time scaling because each module bundles a reusable multi-agent subworkflow (e.g., search + verification). Invoking a module increases reasoning depth at inference time, similar to TTS methods but at the subtask level. Thus, an appropriate collaboration module is invoked for each subtask iteration tasking the budget into consideration.
>
> > **Question 3**: How does the proposed algorithm ensure performance maximization in its design? Are there explicit optimization objectives or adaptive mechanisms?
>
> We do not claim any guarantee of optimal performance. Optimal planning is extremely difficult in our setting because decisions must be made speculatively under uncertain future trajectories. However, as an entirely training-free method, AGENT* leverages test-time scaling—via self-consistency in short-term planning and speculative lookahead in long-term planning—to approximate performance maximization as effectively as possible without learning.
>
> > **Question 4**: What is the specific role of the symbol $\kappa$ in the formula at line 171?
>
> $\kappa$ denotes the coordination strategy that specifies how agents interact that was extracted during the self-play reflection. In our paper, examples include sequential patterns (e.g., ensemble_search → critic), parallel search-and-verify patterns, and iterative refinement patterns extracted through self-play reflection.
>
> > **Question 5**: Is the task step number $H$ fixed or dynamically determined? If it is dynamic, how does this affect the overall budget constraint?
>
> The number of steps $H$ is not fixed. We only impose a maximum step cap, so $H$ is dynamic and determined by the needs of each task until either the budget or step limit is reached.
>
> > **Question 6**: Can the proposed method be extended to other agent collaboration ways beyond orchestrator–worker multi-agent system configurations, like debate-based collaboration?
>
> Yes. AGENT* is intentionally designed to be agnostic to the internal structure of collaboration, as long as interactions can be encapsulated into callable modules with known or estimable costs. A debate-style interaction between two or more agents can naturally be wrapped as a collaboration module $m=(S,\kappa)$, where $S$ is the set of debating agents and $\kappa$ encodes the debate protocol (e.g., number of rounds, aggregation rule). Once such a debate module is defined, it can be treated like any other module in AGENT*.

---

> ### Comment · Reviewer_jpH3 · 2025-11-25
>
> Thanks for the authors’ responses. I have reviewed all replies from 1/3 to 3/3. I still have several concerns:
>
> 1. Regarding Weakness 1, even though the baselines I mentioned involve training, they are still comparable. The authors claim the proposed method is training-free, yet it still requires comparable training cost to those baselines. Thus, rejecting the comparison solely based on being training-free is not reasonable.
>
> 2. Regarding Weakness 2 and Question 3, I still do not see a reasonable explanation of how the method aims to maximize performance. The response to Weakness 2 does not directly address my concern, and the response to Question 3 states that the authors are not aiming to maximize performance. This contradicts Line 60 of the paper: “How can multi-agent systems maximize performance under fixed compute budgets while leveraging complementary agents capable of collaboration?” These responses create confusion regarding the performance objective.
>
> 3. For Weakness 3, I still believe the description of the self-play process omits key implementation details. Providing only the prompt is insufficient. The main text should clearly explain the design rationale and demonstrate the controllability of the LLM prompting process, rather than relying solely on LLM-generated content.
>
> 4. For Weakness 4, the scaling behavior under increased budgets (like \\$0.1, \\$0.5, \\$1, \\$5, to \\$10) remains unclear. It is not evident whether the method exhibits a true scaling law, and it is also unclear how the approach extends to small language models.
>
> 5. For Weakness 5, I do not think it is always reasonable to assume the availability of a validation set. Users may ask questions in any form and across domains, so developers cannot know the question types or the corresponding validation data in advance.
>
> 6. For Question 2, I think the paper lacks a clear scaling factor, such as N in best-of-N or the number of refinement steps in self-refinement. The proposed method does not appear to scale meaningfully with increased budget, and once the budget exceeds a certain value, the scaling effect seems to disappear.

---

> > ### Author Response · Authors · 2025-12-03
> > **Additional Clarifications to Reviewer jpH3**
> >
> > We thank the reviewer again for the careful follow-up reading. We address each remaining point below.
> >
> > > **1. Regarding Weakness 1**
> >
> > Our intention is not to reject comparison on the basis of being “training-free.” Rather, our point is that methods such as GPTSwarm, MaAS, or offline workflow-optimization baselines focus on training-time optimization, which is **orthogonal to our method** which focuses on test-time scaling with a fixed budget constraint.
> >
> > > **2. Regarding Weakness 2 & Question 3**
> >
> > Line 60 states our research question, not an oracle objective. AGENT* aims to **approximate** performance-maximizing behavior under a fixed budget through two mechanisms. The goal is to allocate budget more effectively than existing inference-time baselines, which is reflected in Table 1 and 2’s consistent improvements.
> >
> > > **3. Regarding Weakness 3**
> >
> > We agree more explanation helps. The rationale behind the iteratively prompting LLM to extract the useful collaboration patterns is that **recurring patterns across multiple trajectories indicate stable and genuinely useful coordination behaviors.** We use the default temperature (0.6) for the reflection process prompting without any controlled generation settings.
> >
> > > **4. Regarding Weakness 4**
> >
> > **Our aim is not to claim a law, but to examine compute utilization under practical budgets.** Figure 2 shows AGENT* consistently uses more available budget and converts it to accuracy gains, whereas baselines plateau early.
> > Extending to extremely large budgets ($1–$10) or very small LLMs is an interesting direction, and we will add this explicitly as future work. Our method is orthogonal to model size and can wrap smaller agents, but full evaluation across sizes is outside our current scope.
> >
> > > **5. Regarding Weakness 5**
> >
> > We clarify that the validation set is small, fixed (30 questions), and used only to estimate token costs and discover reusable modules. **It does not require domain-specific labels or task-type curation.** Cost statistics are stable across tasks because the same agents/modules are invoked.
> >
> > > **6. Regarding Question 2**
> >
> > AGENT* scales via two mechanisms: 1) increasing candidate expansion in short-term planning, and 2) deeper speculative rollouts in long-term planning. As shown in Table 1 and Table 2, accuracy continues to improve at higher budgets until the benchmark tasks saturate. While this is not a monotonic “law,” AGENT* exhibits subtask-level scaling, as collaboration modules naturally increase compute-intensive coordination when more budget is available.

---

### Author Response · Authors · 2025-11-22
**Thanks for the reviews**

We sincerely thank all reviewers for their thoughtful feedback and constructive suggestions. Your comments helped us significantly strengthen the clarity, completeness, and empirical rigor of the paper. Below, we summarize the major changes made in our draft to address your reviews. Changes are marked as red in the new draft.

## 1. Defined a new metric, Acc@B (lines 316-320)
We added a clear definition of Acc@B, which measures accuracy under a fixed compute budget B. This metric is now consistently used in all evaluations to enable clearer comparisons.
## 2. Added new baselines (lines 355-362)
Added a new baseline category called “Fixed agent workflows” and added **ADAS** [1] and **AFlow** [2] into the baseline.

[1] Hu et al, Automated design of agentic systems. ICLR 2025 \
[2] Zhang et al, Aflow: Automating agentic workflow generation. ICLR 2025

## 3. Reorganized the main results table (Table 1)
Baselines are grouped into clearer categories:
 (1) Fixed agent workflows
 (2) ReAct with test-time scaling
 (3) ReAct with collaboration modules
This restructuring highlights the gap between approaches and clarifies where AGENT* provides improvements.
## 4. Added comprehensive ablation studies isolating each component of AGENT* (section 4.4, Table 2)
We provide ablations for short-term planning only, long-term planning only, and the full dual-level planner, evaluated under both \$0.2 and \$0.5 budgets.
## 5. Unit of cost and budget (lines 308-309)
We have clarified how the cost and budget are calculated.
## 6. More recent related works (lines 442-448, lines 466-469)
Added recent works on budget-constraint test-time scaling methods and recent multi-agent systems works.

---

### Author Response · Authors · 2025-11-26
**Additional Benchmark on Coding Domain**

To better address reviewer 9zKN and cyXt's concerns on the limited domains of the evaluation benchmarks, **we conducted an additional experiment on BigCodeBench to validate our findings beyond the GAIA and BrowseComp-Plus benchmarks**. BigCodeBench is a challenging coding benchmark consisting of HumanEval-style function-level generation tasks, but with substantially more complex instructions for function completion. We utilize the Instruct subset of the dataset, which consist of code generation tasks based on natural language instructions. In this setting, we define two agents: a coding agent and a verifier agent. Given the requirements, the coding agent produces code, while the verifier agent generates and executes test cases to evaluate the correctness of the generated code. By running the self-play reflection, we curated three collaboration modules:

(1) code_then_verify: a code agent generates the code solution and the downstream verifier agent outputs the result of the verification. \
(2) ensemble_code_then_verify: three code agents generate code in parallel and the verifier agent verifies each code and selects the best code that will fulfill the requirements. \
(3) iterative_code_then_verify: initially run code_then_verify, then based on the verification, the code agent regenerates the code. We fix the iteration by 2.

We have experimented on three core baselines: (1) AFlow, (2) ReAct, (3) ReAct with test-time scaling using two budget constraints: \$ 0.025 and \$ 0.1. Below are the results. For all the agents, we use Claude-3.5-Haiku.

|                   | Acc@0.025 | Acc@0.1 |
| ----------------- | --------- | ------- |
| AFlow             | 46.8      | 46.8    |
| ReAct             | 46.1      | 46.2    |
| ReAct + Best-of-N | 46.1      | 46.2    |
| AGENT*            | 47        | 48.2    |

Overall, AGENT* outperforms all baselines, including the fixed-workflow baseline, ReAct, and ReAct with test-time scaling. Unlike the GAIA and BrowseComp-Plus benchmarks, AFlow exhibits competitive performance on BigCodeBench, even surpassing the ReAct baseline. This suggests that workflow optimization is particularly effective for coding tasks that can be solved in a largely end-to-end manner, in contrast to GAIA-style tasks that involve highly dynamic intermediate decision steps. Meanwhile, ReAct + Best-of-N fails to demonstrate meaningful gains from increased budget. In contrast, **our method shows consistent test-time scaling behavior, where additional budget reliably translates into higher performance.**

Notably, the highest score on the public leaderboard is 51.1, and scores in this benchmark are densely clustered, reflecting its difficulty. As a result, even 1–2 point improvements are considered meaningful. (https://bigcode-bench.github.io/)

---

### Author Response · Authors · 2025-12-03
**Rebuttal Summary of Sumbission 21185**

Dear Area Chair,

Thank you for taking over the evaluation of our submission. We are writing to summarize the full rebuttal process.

# Clarification on Score Changes and Response Timeline
Our submission initially received **4, 4, 4, 6**. After rebuttal, but before the API leakage event, the scores became **4, 4, 6, 6**. We submitted all rebuttal responses on Nov 22 PST. Although **9zKN** (4) did not reply, three reviewers responded before the incident.

Three reviewers responded during the discussion phase:
- cyXt (4 → 6) acknowledges that our clarifications resolve their key concerns and raises the score on Nov 23, 05:04 PST.
- jpH3 (4) maintains some concerns, though all factual clarifications were addressed on Nov 25, 15:13 PST.
- XHcw (6) indicates that our rebuttal addresses most concerns and maintains a positive assessment on Nov 25, 18:53 PST.

According to the OpenReview statement, the ICLR 2026 Workflow Chairs first reported the bug on Nov 27, 07:09 PST, after which large-scale exploitation began. All reviewer responses and score changes therefore occur well before this time.

# Positive Aspects Highlighted by Reviewers
- **Clear and timely problem motivation.**
jpH3, 9zKN, XHcw, and cyXt emphasize that budget-aware multi-agent compute allocation is an important and novel problem setting.
- **Intuitive and well-structured framework.**
XHcw and cyXt praise the clarity of AGENT*’s design, noting that the dual-level planner and collaboration modules are coherent and intuitive.
- **Consistent and meaningful empirical gains.**
jpH3, 9zKN, and cyXt highlight AGENT*’s reliable improvements across budgets, models, and tasks.
- **Thorough experimentation and analysis.**
XHcw and cyXt view the evaluations on GAIA and BrowseComp-Plus as systematic, sound, and demonstrative of effective budget use.

# Key Concerns Raised and Our Responses

## 1. Reviewer jpH3 (4)
- **Insufficient baseline design.**
We added ADAS and AFlow as new baselines (**lines 355-362, Table 1**). We also justify that standard TTS baselines (Best-of-N, Iterative Verification) are the most widely adopted baselines.
- **“Performance maximization” objective.**
We clarified that AGENT* approximates performance-maximizing behavior via short-term evaluation and long-term budget feasibility, without claiming global optimality.
- **Scaling behavior & model-size effects.**
We clarified that AGENT* is orthogonal to parameter scaling. Experiments already involve models with diverse token costs and capabilities.
- **Validation set realism.**
We argued that a small validation set is a realistic assumption for training-free systems and is used only to estimate cost profiles and induce reusable modules.

Reviewer jpH3 maintained some concerns post-rebuttal, though all factual clarifications were addressed.

## 2. Reviewer 9zKN (4, no post-rebuttal response)
- **Magnitude and statistical meaning of improvements.**
We clarified that AGENT* yields substantially larger improvements over the strongest external baseline (ReAct + Iterative Verification).
- **Limited benchmarks.**
GAIA and BrowseComp-Plus were chosen as they uniquely require heterogeneous multi-agent collaboration. Additionally, we conducted new experiments on BigCodeBench, confirming similar improvement trends in coding tasks.
- **Missing component-wise ablations.**
We added an ablation study (**section 4.4, Table 2**) isolating short-term planning, long-term planning, and their combination, demonstrating that both components contribute and perform best.
- **Missing recent workflow optimization work.**
We added MaaS and FlowReasoner to related work (**lines 442-448, lines 466-469**) and incorporated ADAS and AFlow as new baselines in Table 1.
- **Budget definition & reproducibility.**
We clarified that the “budget” is computed directly from token pricing for each model (**lines 308-309**), making the cost calculation transparent, model-agnostic, and reproducible on any API.
- **Presentation issues.**
We reorganized baselines into clearer groups, introduced Acc@B for readability (**lines 316-320**), and improved figure and table.

Reviewer 9zKN did not respond, but we believe we have addressed all the concerns.

## 3. Reviewer XHcw (6, keeps positive score)
- **Missing related work on budget-aware agentic reasoning.**
Same as 9zKN
- **Need for stronger multi-agent baselines.**
same as jpH3

Reviewer XHcw confirmed that the rebuttal resolved most concerns.

## 4. Reviewer cyXt (4->6)
- **Limited benchmarks.**
Same as 9zKN.
- **Significance and variability of improvements.**
Same as 9zKN.
- **Budget definition & reproducibility.**
Same as 9zKN.
- **Ablation of planning components.**
Same as 9zKN.

Reviewer cyXt acknowledged that concerns were resolved and raised the score from 4 to 6. Also, we have added BigCodeBench coding benchmark to further address the remaining concern about the limited benchmark coverage.

---

### Meta-Review · Area_Chair_AveN · 2026-01-07

**Summary:**

The paper proposes AGENT*, a framework designed for budget-aware test-time scaling (TTS) in multi-agent systems (MAS).
It introduces a dual-level planner and reusable collaboration modules to optimize compute allocation.
While all reviewers recognized the importance and novelty of the problem setting, the decision to reject is primarily informed by unresolved concerns regarding the methodological rigor.
Specifically, the reviewers questioned the conceptual boundary of the "training-free" claim, given the method's reliance on a validation set for module discovery.
Furthermore, the empirical evidence for consistent scaling behavior across diverse budgets was deemed insufficient.

**Reviewer Concerns:**

Addressed: During the rebuttal, the authors successfully addressed concerns regarding benchmark diversity by adding results on BigCodeBench,
showing that AGENT* generalizes beyond web-navigation tasks. They also clarified the definition of "budget" based on token pricing, which resolved reproducibility issues.
Presentation improvements (clarifying tables and figures) were also acknowledged.

Outstanding: The most critical concern remains the methodological dependence on a validation set.
Reviewer jpH3 (Confidence 5) noted that using a validation set for module discovery makes the system comparable to MAS optimization frameworks that were not included as baselines.
Additionally, the theoretical justification for "performance maximization" and the empirical scaling trends at higher budgets remain weakly supported, with reviewers noting that performance gains are modest and sometimes plateau inconsistently.

**Reviewer Scores:**

jpH3 (4): Likely would have maintained a score of 4. Although the reviewer acknowledged the rebuttal, they remained firm on the technical flaws regarding the validation set and the lack of explicit optimization objectives.
9zKN (4): Likely would have remained at 4. This reviewer pointed out that improvements were modest and lacked statistical validation, and they did not update their score post-rebuttal.
XHcw (6): Participated in the discussion and maintained a 6, reflecting satisfaction with the rebuttal's clarifications on budget-aware planning.
cyXt (6): Officially upgraded from 4 to 6 after the rebuttal, satisfied with the cost mapping and the additional coding benchmarks.

---

### Decision · Program_Chairs · 2026-01-26

Reject